# Solution-Processed Functionalized MoS$_2$ Nanosheets Composite for Photodetection Application

Alexander V. Kukhta [1,*,†], Enliu Hong [2,†], Nadzeya I. Valynets [1], Sergei A. Maksimenko [1], Uladzislau Parkhomenka [3], Nikita Belko [3], Anatoly Lugovsky [3], Tatiana A. Pavich [4], Iryna N. Kukhta [5], Ziqing Li [2] and Xiaosheng Fang [2]

[1] Institute for Nuclear Problems, Belarusian State University, 220006 Minsk, Belarus; valynets@inp.bsu.by (N.I.V.); inp-director@inp.bsu.by (S.A.M.)
[2] Department of Materials Science, Institute of Optoelectronics, Fudan University, Shanghai 200437, China; 21110300005@m.fudan.edu.cn (E.H.); lzq@fudan.edu.cn (Z.L.); xshfang@fudan.edu.cn (X.F.)
[3] A.N. Sevchenko Institute for Applied Physical Problems, Belarusian State University, 220006 Minsk, Belarus; parkhomm2@gmail.com (U.P.); belkonv@bsu.by (N.B.); an.lugovsky@yandex.ru (A.L.)
[4] B.I. Stepanov Institute of Physics, National Academy of Sciences of Belarus, 220072 Minsk, Belarus; pavich56@mail.ru
[5] Institute of Chemistry of New Materials, National Academy of Sciences of Belarus, 220072 Minsk, Belarus; ir.kukhta@gmail.com
* Correspondence: kukhta@bsu.by or al.kukhta@gmail.com
† These authors contributed equally to this work.

**Abstract:** Charge-transfer organic-inorganic complexes have demonstrated great potential in optoelectronic applications. Herein, a drop-casting processed photodetector based on thick composite films made of multi-layered MoS$_2$ nanosheets chemically bonded to linear molecules of aromatic thiols has been developed. Composites based on multilayered nanosheets allow for facile preparation of low-cost, large-area, and flexible devices. It was demonstrated that a simple functionalization of ultradispersed MoS$_2$ nanosheets with linear aromatic thiol results in the formation of charge and energy transfer complexes. A photodetector with functionalized MoS$_2$ nanosheet film prepared by drop coating with Au electrodes demonstrated enhanced performance compared to pure materials. Our first experiments illustrated that functionalization of MoS$_2$ nanosheets by a paraquaterphenyl thiol derivative leads to a significant increase in the photoresponse speed (by a factor of 12) and decay speed (by a factor of 17.5), in addition to the enhancement of the photostability of the MoS$_2$ based photodetector. The photo current value has been increased by about an order of magnitude. The proposed approach offers promising prospects for further development of photodetectors.

**Keywords:** 2D material; MoS$_2$; aromatic thiol; optical properties; photodetector

## 1. Introduction

Photodetectors are essential for detecting and measuring light intensity using the function of converting optical signals to electrical signals. Photodetectors offer manifold various applications [1]. New materials and structures for photodetectors are constantly being developed due to the evolving applications and updated requirements. Two-D materials (such as graphene, transition metal dichalcogenides, etc.) have been ubiquitously used as the most promising semiconductor materials for this purpose [2,3]. Molybdenum disulfide (MoS$_2$) nanosheets are among the most studied dichalcogenides [4,5]. This material has attracted more academic and applied interest due to its extraordinary properties arising from the internal structure of the nanosheets with d-electron interactions and its great degree of tunability. A single-layer MoS$_2$ nanosheet forms a hexagonal lattice structure and contains three sublayers of atoms, where one layer of Mo atoms is sandwiched between two layers of S atoms. Most photodetector studies [6–17] use single-layered and multilayered individual nanosheets, though positive results with multilayered nanosheet composites

in the sandwich structure were shown [18]. Single-layered nanosheets synthesized by CVD are widely known to be very time and energy-consuming and, therefore, expensive. Mechanically exfoliated nanosheets can be produced simply, but they have limitations in mass production. Multilayered $MoS_2$ nanosheets for composite formation can be easily obtained by cheap, simple, and reliable liquid phase exfoliation [4]. Composites based on multilayered nanosheets can be solution-processed, which allows for facile preparation of low-cost, large-area, and flexible devices [19]. Unfortunately, the characteristics of these photodetectors are insufficiently high because of a lower ordering degree than the one in independent single nanosheets, as well as the presence of contacts between the nanosheets, resulting in lower conductivity. To achieve good performance of photodetectors (high responsibility, short response time, and high detectivity), both the physical and chemical functionalization of such nanosheets are widely utilized [3,13–17,20,21]. Many inorganic [1,3,12] and organic [3,8,9,13–17,20,21] nanomaterials have been used for functionalization. Most of the studies are devoted to physical functionalization [14–17,21]. However, chemical functionalization can provide more stable materials as it is irreversible. A wide variety of organic materials with different properties gives wide possibilities for functionalization. Promising results have been obtained with aliphatic thiols [11]. However, aromatic organic materials are electrically conductive and absorb light more strongly. Thus, in this study, we propose to apply the chemical functionalization of $MoS_2$ multilayered nanosheets with a linear aromatic thiol, which is stable and luminescent. A combination of a *n*-type $MoS_2$ nanosheet and a *p*-type aromatic molecule can form a charge transfer complex [22]. Often, high gain can be reached owing to the photogating effect [23,24], capable of achieving higher characteristics and escaping a cooling system, lowering the detector's noise. Chemical functionalization of $MoS_2$ nanosheets by organic molecules can be considered as covering by surfactant, which prevents nanosheets from aggregation [25], maintains stable interaction between nanosheets, and changes the properties of these nanosheets [26,27], which can be used for the creation of stable composite films. Thus, in this work, we propose to develop solution-processed films based on the newly synthesized material and study their photodetecting properties. Our approach results in the formation of a high-quality film, the enhancement of the photoresponse value and speed, and the photostability of the $MoS_2$-based photodetector. Due to the simplicity of chemical functionalization, our proposed approach is promising for the further development of solution-processable photodetectors.

## 2. Materials and Methods

### 2.1. Materials

The $MoS_2$ ultrafine powder with hexagonal lamellar structure was purchased from Graphene Supermarket (~90 nm average particle size, purity 99%, product number EM092222) and used as supplied. Functionalized $MoS_2$ ($MoS_2$-M) has been synthesized using this powder. Chlorobenzene (99.5%, CB) and isopropanol (99.8%) were used as received without further purification. $MoS_2$ nanosheets were exfoliated in isopropanol based on a widely used exfoliation technique [4] with an ultrasonic tip for 12 h, resulting in the formation of a dispersion of $MoS_2$ nanosheets with a wide range of sizes and thicknesses. After sonication, the dispersion has been used for optical measurements and film preparation.

### 2.2. MoS₂ Nanosheets Functionalization

For the functionalization of the $MoS_2$ nanosheets, 4-Hexyl-3′-methyl-4‴-mercapto-*p*-quaterphenyl (MQP) has been chosen. The chemical structure of MQP is presented in Figure 1.

**Figure 1.** Chemical structure of MQP molecule.

The process of the chemical functionalization of MoS$_2$ nanosheets with thiol molecules is simple and reliable, as presented in detail herein. Ultrasonic (US) disaggregation of the MoS$_2$ ultrafine powder suspension in an organic solvent with the addition of MQP to maximize the MoS$_2$ specific area and the MQP chemisorption has been applied. For this purpose, the following materials and apparatus were used: MoS$_2$ ultrafine powder, DMF and *n*-hexane from Merck (Germany); MQP (synthesized); ultrasonic cleaner GT Sonic 1860QT (150 W, 40 kHz); centrifuge MDCEN-302-SD; diaphragm pump Wiggens C610; UV-lamp with 264 and 365 nm sources; glass round-bottom flasks.

A 50 mg MoS$_2$ (MW 160 g/mol; 0.3 mmol) suspension in DMF (10 mL) was treated by US for 15 min in a round-bottom flask. After dispersion and continuing US treatment, a solution of 140 mg MQP (MW 436 g/mol; 0.3 mmol) in DMF (5 mL) was added with a rate of about one drop/s. Then, the reaction mixture was sonicated again for 1 h. The obtained suspension was evaporated and dried under vacuum (15 mm Hg, 50 °C, 30 min). The residue was dispersed in *n*-hexane (20 mL) under US to remove the excess MQP. This mixture was next separated in a centrifuge (6000 RPM, 5100 g, 30 min); the residue was rinsed with hexane (five times) until the solvent after centrifugation showed no fluorescence due to the traces of MQP when excited with a 365 nm UV light. After the purification, a wet material was dried on air at 50 °C.

MQP was synthesized specifically for this work, as it is not commercially available. The synthesis of MQP was carried out by the interaction of 4-hexyl-3′-methyl-4′′′-Br-*p*-quaterphenyl (II) with *n*-butyllithium at a temperature ranging from −70 to −78 °C, followed by the addition of elemental sulfur. Next, 4-hexyl-3′-methyl-4′′′-Br-*p*-quaterphenyl was obtained by the Suzuki reaction of 4-hexyl-3′-methyl-4′′′-B(OH)$_2$-*p*-terphenyl (I) with 4-bromoiodobenzene (see Scheme 1).

**Scheme 1.** Synthetic route 4-hexyl-3′-methyl-4′′′-mercapto-*p*-quaterphenyl (MQP).

The 4-Hexyl-3′-methyl-4′′′-bromo-*p*-quaterphenyl (II) (4,83 g, 10 mmol) was dissolved in 50 mL of dry THF, and the solution was cooled in an argon atmosphere to −80–70 °C with continuous stirring. At this temperature, an *n*-butyllithium solution (5 mL of 2.5M solution in hexane) was added by syringe, and the reaction mixture was stirred for 1 h at the same temperature. The sulfur solution (0.384 g, 12 mmol) in 30 mL THF was added to this mixture by syringe. The mixture was gradually allowed to warm up to 20 °C, followed by the addition of 50 mL of 2% hydrochloric acid. The organic layer was separated, and the aqueous layer was extracted with two 30 mL portions of toluene. The combined organic

layer was washed with water twice and dried over the anhydrous magnesium sulfate. After filtration, the solvent was removed in a vacuum to yield 4.22 g of crude 4-hexyl-3′-methyl-4‴-mercapto-*p*-quaterphenyl. The crude product was recrystallized from the acetonitrile/acetone (3/1) solution. The yield of MQP was 3.76 g (86%). The $^1$H NMR (400 MHz, CDCl$_3$, δ, ppm) was 7.85 (m, 1 H), 7.73~7.63 (m, 5 H), 7.25~7.45 (m, 10 H), 7.06 (m, 1 H), 3,40 SH, 2.77~2.62 (m, 2 H), 2.59 (s,3 H), 1.59 (dd, *J* = 14.6, 7.0 Hz, 2 H), 1.29 (dd, *J* = 8.6, 5.3 Hz, 4 H), and 0.91 (*t*, *J* = 6.6 Hz, 3 H).

Compound II has been synthesized following the procedure described below. The mixture of 4-bromoiodobezene 4.53 g (16 mmol), 4-hexyl-3′-methyl-*p*-terphenyl-4‴-boronic acid (I) 4.67 g (12,5 mmol), potassium carbonate 4.2 g (30 mmol), acetone (60 mL), THF (20 mL), and water (40 mL) was refluxed upon continuous stirring for 0.5 h in an argon atmosphere, followed by the addition of palladium acetate (20 mg). The reaction mixture was refluxed under stirring for 3 h. Upon the evaporation of acetone, 1N solution hydrochloric acid was added at pH = 2. The crude product was extracted with methylene chloride, washed with water, and dried over anhydrous magnesium sulfate (MgSO$_4$). The crude product was then purified by column chromatography (using silica gel, methylene chloride as eluent). The solvent was removed using a rotary evaporator, and the product was recrystallized from acetonitrile/acetone (3/1). The yield of II was 5.41 g (89%): $^1$H NMR (400 MHz, CDCl$_3$, δ, ppm): 7.85 (m, 1 H), 7.73~7.66 (m, 5 H), 7.25~7.45 (m, 10 H), 7.06 (m, 1 H), 2.77~2.62 (m, 2 H), 2.59 (s,3 H), 1.59 (dd, *J* = 14.6, 7.0 Hz, 2 H), 1.29 (dd, *J* = 8.6, 5.3 Hz, 4 H), 0.91 (*t*, *J* = 6.6 Hz, 3 H); MS(ESI): 484,482 (M$^+$), 413,411.

The synthesis of 4-hexyl-3′-methyl-4‴-B(OH)$_2$-*p*-terphenyl (I) was carried out following the procedure described in the literature [28].

MoS$_2$ nanosheets can be presented by several polymorphs, depending on the interlayer stacking arrangement and intralayer coordination between the central Mo atom and the surrounding S atoms [3–5]. The exact nature of organic thiol and exfoliated MoS$_2$ interactions remains unclear [29]. Thus, it is not clear where the thiol group of the molecule can be bonded to the MoS$_2$ nanosheet. Typically, the defect-free area of a MoS$_2$ basal plane is inert. However, it was observed that an S of thiol can be attached to the Mo [30,31] or S [30,32,33] atoms at vacancy defects. Mo atoms can be reached at the edges of the nanosheet, while S atoms can be accessible at the basal planes.

### 2.3. Thin Film Preparation

Thin films were obtained by drop-casting the obtained suspension on the substrate or by spin-coating. For thin film deposition, we used either 0.5 mm thick quartz substrates transparent for wavelengths above 200 nm or silicon surface for device preparation. All substrates were carefully cleaned with acetone and ethanol and were dried before thin-film deposition.

### 2.4. Device Fabrication

First, the sample powder was added to the CB solvent and sonicated for 12 h to form a dispersion. After that, uniform films were cast on a clean SiO$_2$/Si substrate by drop/spin coating, followed by annealing at a specific temperature for 1 h. To ensure the cleanliness of the substrate, it is necessary to perform ultrasonic cleaning treatment on the substrates before usage. Specifically, the SiO$_2$/Si substrates are cleaned with acetone and isopropyl alcohol and deionized water for 30 min. Acetone and alcohol can remove the majority of organic impurities from the substrates' surface, and deionized water can wash away the remaining particle contaminants. To improve film homogeneity, the samples need to be annealed. However, because regular annealing temperatures for inorganic materials [34] are too high for organic materials, we lowered the annealing temperature and shortened the annealing time for composite samples to 80 °C for 1 h. Unfortunately, such a low temperature cannot remove all defects. The thickness of the active layer film for both devices is about 1.3 μm (1.32 μm for MoS$_2$ and 1.28 μm for MoS$_2$-M). After optimizing synthesis parameters and obtaining uniform, high-quality films, Au electrodes (70 nm

thick) were plated on the film by thermal vacuum evaporation. As a result, two kinds of photodetectors (PDs) based on pure $MoS_2$ and doped $MoS_2$ were formed.

### 2.5. Characterization

Systematic characterizations were carried out to determine the properties of functionalized $MoS_2$ nanosheets and their individual counterparts.

Absorption spectra were recorded with a Cary 500 (Varian, Palo Alto, CA, USA) spectrophotometer. Photoluminescence and excitation spectra were measured with a multifunctional spectral fluorimeter, Fluorolog-3 (Horiba Scientific, Palaiseau, France), which provides highly sensitive and stable measurements in the ultra-violet and visible range. All photoluminescence measurements for solutions were made perpendicularly to the excitation beam, and thin films were oriented at 30°. Raman spectra have been obtained at room temperature with a micro-Raman spectrometer Nanofinder High End (Lotis TII, Minsk, Belarus—Tokio Instruments, Tokio, Japan; 1800 lines/mm grating; 50× objective; 532 nm excitation wavelength). The morphology of the dispersed phase was examined using a field emission scanning electron microscope (FESEM), Zeiss Sigma, and scanning electron microscope MIRA3 (Tescan, Brno, Chech Republic). The thickness of the film is determined by a step profilometer (KLA-Tencor, Milpitas, CA, USA). All measurements were carried out at room temperature and ambient conditions. The molecular structure of the synthesized organic compounds was verified using [1]H NMR spectra acquired on a Bruker AVANCE 500 spectrometer at room temperature and mass spectra acquired on an Agilent 6410 Triple Quadrupole LC/MS System.

Optoelectronic properties were collected using the semiconductor characterization system Keithley 4200-SCS (Tektronix, Beaverton, OR, USA) connected to a vacuum probe station (Lake Shore). A 75 W Xe lamp equipped with a monochromator was used as a light source. Light density was measured by a NOVA II power meter (OPHIR photonics, Jerusalem, Israel). All the measurements were performed at ambient environment, i.e., room temperature and atmospheric pressure.

## 3. Results and Discussion

### 3.1. Microscopy

Electron microscopy provides insights into the morphology of the films. Figure 2 represents an SEM picture of the morphology of $MoS_2$ and $MoS_2-M$ films. It can be seen that this sample consists of large nanosheets (of several micrometers) covered with small particles, hinting that US treatment was not complete, especially for pure $MoS_2$. Dispersion size and concentration are known to depend on sonication time and power [35]. The plane of the large nanosheets is located mainly parallel to the plane of the substrate surface, and the small nanosheets are positioned similarly on the surface of the large nanosheets (see Figure 2, right). Additionally, no fundamental difference between $MoS_2$ and $MoS_2$-M film morphology has been observed. Hence, only a small part of the nanosheet surface has been covered with molecules that are typically oriented perpendicularly to its surface. The combination of large and small particles can result in higher conductivity owing to the denser packing of the particles. Note that the low-resolution microscopy reveals a much lower quality of the pure $MoS_2$ nanosheet film than the film made of $MoS_2$-M nanosheets. $MoS_2$ film samples contain numerous inhomogeneities and hollow cavities. Functionalized $MoS_2$ nanosheet films are much more homogeneous than nonfunctionalized material because of better interaction between the nanosheets caused by thiol molecules.

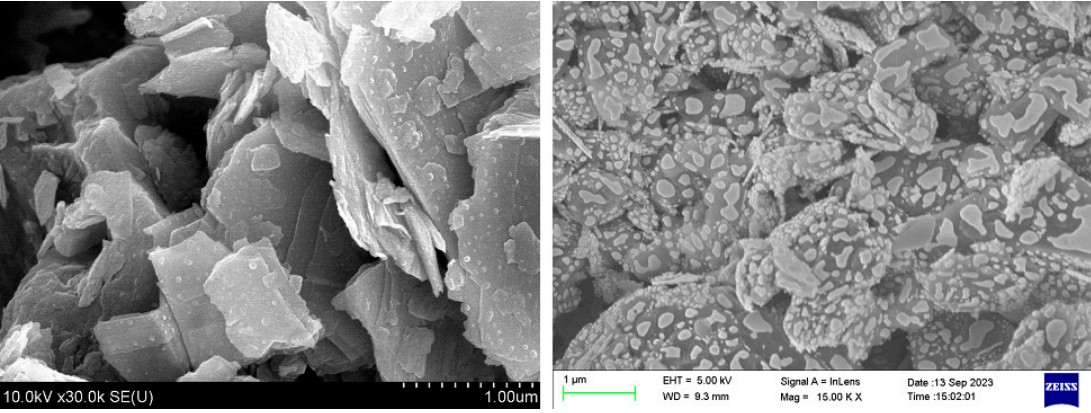

**Figure 2.** FESEM scans of the $MoS_2$ (**left**) and the $MoS_2-M$ (**right**) thin film samples.

### 3.2. Raman Spectra

Raman spectroscopy is known to be a reliable diagnostic tool for studying the thickness of $MoS_2$ nanosheets, as well as various interactions [36]. Raman spectra of unsonicated $MoS_2$ and functionalized $MoS_2-M$ samples (Figure 3) excited with a 532 nm laser were obtained. Both spectra contain two main modes of out-of-plane vibrations ($A_{1g}$) of S atoms and in-plane vibrations ($E^1_{2g}$) of Mo atoms and S atoms. The original $MoS_2$ nanosheets show these peaks at about 381 and 407 $cm^{-1}$, with the difference between them of ~26 $cm^{-1}$ implying that this sample contains nanosheets with thicknesses ranging from monolayer to, preferably, several layers. Functionalized $MoS_2$ nanosheets show a negligible (about 1 $cm^{-1}$) redshift of both peaks with the same peak intensity ratio but a noticeable broadening of both peaks. These data confirm the presence of both S−S and S−Mo bonds owing to the successful $MoS_2$ functionalization. Note that the measurements were made at five points with different nanosheet dimensions, distribution, and morphology for both samples. It was shown that the only observed difference was the signal intensity.

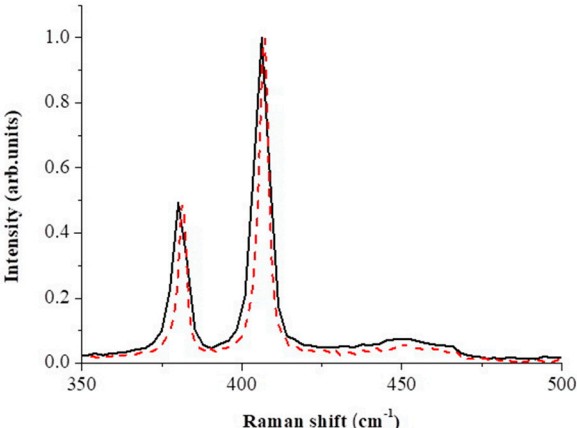

**Figure 3.** Normalized Raman spectra of $MoS_2-M$ (solid) and $MoS_2$ (dashed) thin films.

### 3.3. Absorption and Luminescence

Optical absorption and luminescence spectra can give information about the position of energy levels and energy transformation in the considered system. Figure 4 illustrates the normalized absorption, excitation, and luminescence spectra of $MoS_2-M$ and the corresponding components ($MoS_2$ and MQP). The absorption spectrum of $MoS_2-M$ is approximately a superposition of the absorption of both individual components. However, a longwave shoulder at about 350 nm can be considered a charge transfer band. Indeed, the relative intensity of this shoulder is much higher than that of the $MoS_2$ spectrum, though the deposition of MQP is negligible. Partial overlapping of $MoS_2$ absorption and MQP

luminescence spectra can result in the partial energy transfer from MQP to MoS$_2$. Blue luminescence of MoS$_2$ occurs, probably, from higher energy levels as described in [37–40]. It is weak compared to the luminescence of MQP and the composite. No red luminescence is observed in the system of few-layered MoS$_2$ nanosheets. The luminescence spectrum of the hybrid sample is red-shifted by about 19 nm and broadened compared to the MQP spectrum because of the partial superposition of the luminescence spectra of both counterparts. The intensity of its luminescence is much weaker than that of MQP. This finding indicates that a charge transfer from MQP to MoS$_2$ occurs, which is confirmed by the almost fully coinciding excitation spectra of MQP and MoS$_2$-M. Thus, in this system, both energy and charge transfer can be observed.

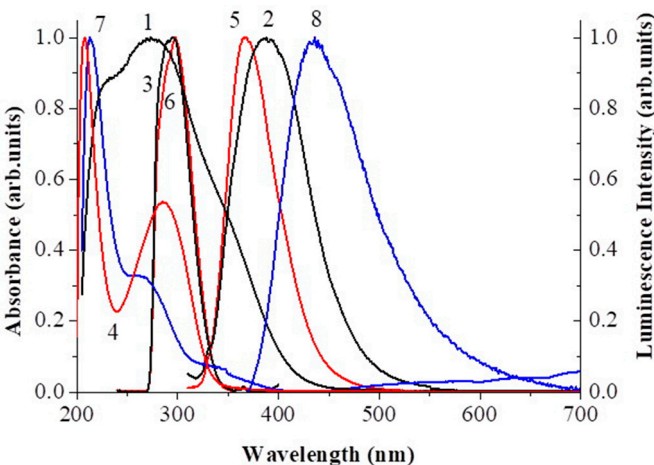

**Figure 4.** Normalized absorption (1,4,7), luminescence (2,5,8), and luminescence excitation (3,6) spectra of MoS$_2-$M (1–3), MQP (4–6), and MoS$_2$ (7,8) isopropanol solutions.

### 3.4. Current-Voltage Characteristics of Photodetectors

Subsequently, the optoelectronic performance of the constructed devices was tested. The length of the devices' channel is 100 μm, and the width of the gold electrodes is 500 μm. The thickness of SiO$_2$ is 300 nm, which is sufficient to eliminate the interference of the bottom substrate (silicon) signal. As seen from Figure 5a,b, the structures of the devices based on pure MoS$_2$ and MoS$_2-$M are similar, with the only difference in the composition of the photosensitive layer with the same thickness of about 1.3 μm. As a result, as shown in Figure 5c,d representing *I-V* test results, there are two obvious improvements in optoelectronic performance after the doping of organic molecules. On the one hand, both dark and photocurrent were increased by about one order of magnitude. On the other hand, the current signals are much more stable than that of pure MoS$_2$. Compared to the pure MoS$_2$ device, the organic molecules$-$doped MoS$_2$ device shows significant positive effects. Before doping, the fabricated device shows obvious current instability, both dark current and photocurrent, indicating that the charge carrier transmission is very inefficient. This may be caused by the high defect density of materials, low carrier mobility, small diffusion length, etc. More importantly, the film quality of pure MoS$_2$ is limited without the MoS$_2$ doping, which will affect the generation, separation, transportation, and recombination of photogenerated carriers to a great degree, thus leading to poor performance with jittering intense current signals and slow response behavior under light illumination. It should be noted that the dark current at negative bias is higher than the photocurrent at different wavelengths of laser irradiation, indicating that the photodetectors do not have any photo gain at negative bias. In contrast, positive bias does not have this problem. The reason for this phenomenon can be the formation of Schottky contacts between metal and semiconductors. In the case of Schottky contact, the band of semiconductor at the interface is curved, thus forming a Schottky barrier and showing a nonlinear *I-V* behavior under positive/negative bias. It is, therefore, rational that the photocurrent is higher than

the dark current at positive bias and lower at negative bias because of the existence of a built-in electric field. This property can be changed with Ohmic contacts in the case of corresponding energy levels of electrode and semiconductor. On the other side, gold electrodes can be replaced by polymer electrodes.

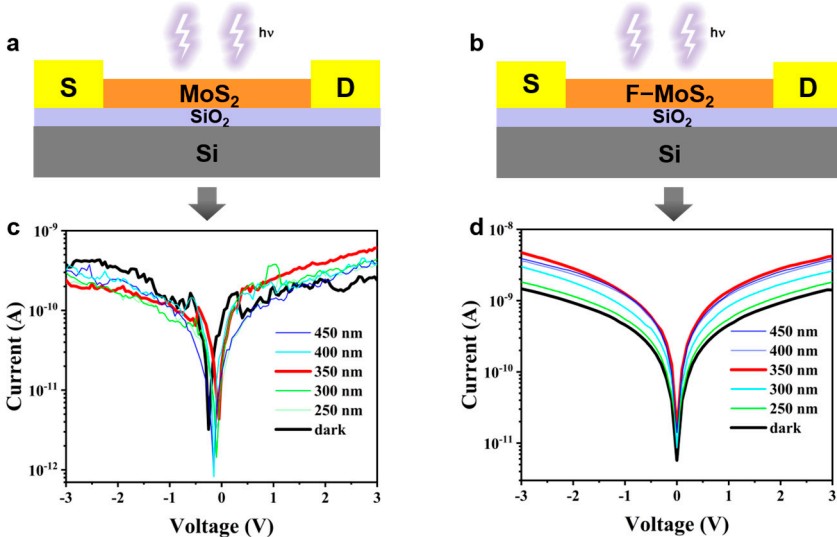

**Figure 5.** (**a**,**b**) The device structures of pure $MoS_2$−based and $MoS_2$−M−based PDs; (**c**) I–V curves of pure $MoS_2$−based PD; and (**d**) I–V curves of doped−$MoS_2$−based PD.

### 3.5. Photoresponse Kinetics

To further study the enhancement of the optoelectronic performance of doped−$MoS_2$-based photodetectors, we performed time-dependent *I-t* tests of the devices. The incident laser power over the fabricated device is 228 µW cm$^{-2}$ (at 350 nm). As seen in Figure 6b, compared to pure $MoS_2$, the $MoS_2$−M−based PD exhibits better properties with higher currents and more stable photocurrents, especially with faster response speed. As specifically shown in Figure 6c,d, the rise $\tau_{rise}$ and decay $\tau_{decay}$ time of pure $MoS_2$ are 5.2 s and 14.8 s, respectively, and the $\tau_{rise}$ and $\tau_{decay}$ time of $MoS_2$−M are 0.4 s and 0.8 s, respectively, which means the response speed is increased by 12 times, and the decay speed is increased by 17.5 times. These rise and decay times originate because of the presence of nanosheet edges acting as traps [17] retarding the movement of charges. Note that there are fast and slow decay components (Figure 6). This tail of the slow decay component can be attributed to the adsorption of molecules of oxygen and water from air [15] and other long-lived traps.

Interestingly, the response speed and photocurrent stability were greatly improved. We attribute these improvements to the good electrical conductivity of organic molecules, which creates better contacts and connections between the nanosheets. This result improves film quality with better morphology, such as better particle distribution, improved uniformity, more substantial filling, etc., that is in consistent with the microscopy data. This factor facilitates the transport of photo-generated carriers and improves charge carrier dynamics and photocurrent stability. Indeed, better contacts between nanosheets in functionalized samples increase the flow of charges as occurs in graphene nanosheets with metal nanoparticles fully covered by aromatic thiol molecules [41], which can be noticed as the increase in response speed and value. Also, light absorption is increased after the $MoS_2$ functionalization due to the additional absorption by the organic molecules. It can be concluded from the above results that doping by organic molecules enhances the photoresponse speed and the photostability of $MoS_2$. Although the device performance is not outstanding compared to the device based on the mechanically exfoliated $MoS_2$, this novel doping strategy of combining organic molecules with transition metal dichalcogenides can result in the realization of stable and accurate control of photo-generated carriers in a composite

film. It has great significance for expanding the application potential of two-dimensional materials in semiconductor optoelectronic devices.

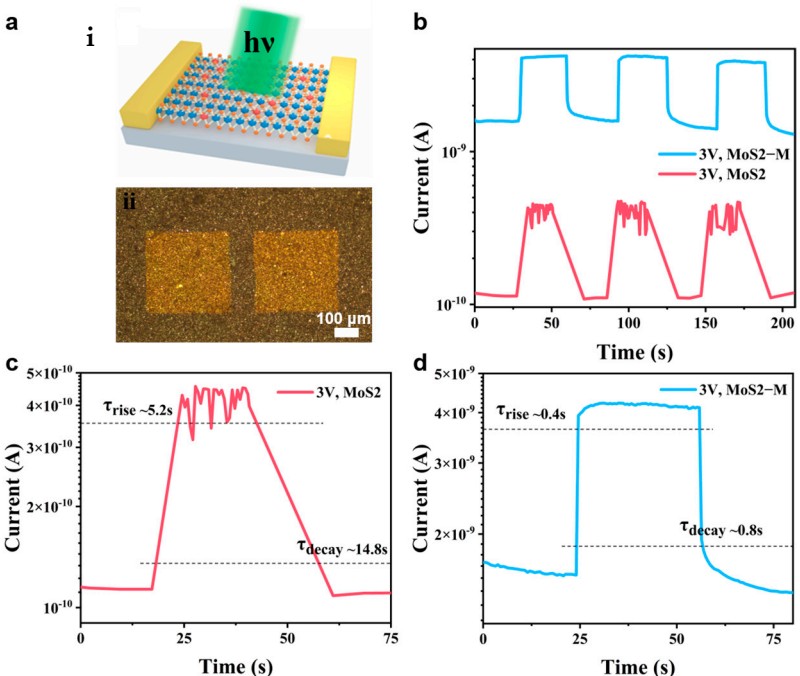

**Figure 6.** (**a**) Schematic illustration of the device architecture (**i**) and an optical microscope image of the real device (**ii**); (**b**) photoresponse comparison of pure $MoS_2$−based and $MoS_2$−M−based PDs; (**c**) semilogarithmic I-t curves of pure $MoS_2$−based PD under 350 nm illumination on/off switching at 3 V; (**d**) semilogarithmic I-t curves of $MoS_2$−M−based PD under 350 nm illumination on/off switching at 3 V.

## 4. Conclusions

The photodetector based on thick composite films made of multi-layered $MoS_2$ nanosheets chemically bonded with linear molecules of aromatic thiols has been developed. Composites based on multilayered nanosheets can be solution-processed, which allows for the simple fabrication of low-cost, large-area, and flexible devices. The sample consists of large nanosheets (of several micrometers) covered with small $MoS_2$ particles. Raman line-broadening and red-shift to lower-wavenumbers of the $MoS_2$ peaks confirm the formation of the functionalized composite. Both absorption and luminescence spectra are the superposition of the corresponding components. However, the long-wavelength shoulder at about 350 nm in the absorption spectrum of the composite can be considered as a charge transfer band. It was found that the functionalization of $MoS_2$ nanosheets with a paraquaterphenyl thiol derivative greatly enhances the photoresponse value, speed, and photostability of $MoS_2$-based photodetector. This study represents our first steps in the proposed direction and requires further experiments. The proposed approach is very promising for further development, for example, the use of higher dispersity, choice of a functionalizing molecule, and degree and type of functionalization.

**Author Contributions:** Conceptualization, A.V.K. and X.F.; methodology, A.V.K. and A.L.; software, N.B.; validation, I.N.K.; formal analysis, I.N.K.; investigation, E.H. and N.B.; resources, U.P. and A.L.; data curation, A.L. and T.A.P.; writing—original draft preparation, A.V.K. and E.H.; writing—review and editing, A.V.K.; visualization, N.I.V.; supervision, S.A.M. and X.F.; project administration, Z.L.; funding acquisition, S.A.M. and X.F. All authors have read and agreed to the published version of the manuscript.

**Funding:** The authors extend their appreciation to the Belarusian Foundation for Fundamental Research (Project No T23KI-004) for funding and supporting this work. Xiaosheng Fang acknowledges the support from National Natural Science Foundation of China (No. 12211530438) and the Office of Global Partnerships (Key Projects Development Fund).

**Institutional Review Board Statement:** Not applicable.

**Informed Consent Statement:** Not applicable.

**Data Availability Statement:** The data that support the findings of this study are available within the article.

**Acknowledgments:** The authors extend their appreciation to the Belarusian Foundation for Fundamental Research for funding and supporting this work. Xiaosheng Fang acknowledges the support from the National Natural Science Foundation of China and the Office of Global Partnerships (Key Projects Development Fund).

**Conflicts of Interest:** The authors declare no conflict of interest.

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
