# Peer review of "Solution-Processed Functionalized MoS2 Nanosheets Composite for Photodetection Application"

_photonics, doi:10.3390/photonics10121295_

Round 1

Reviewer 1 Report

Comments and Suggestions for Authors

In this manuscript, the authors fabricated the photodetectors of MoS2 and functionalized MoS2. The work is interesting but needs major revision before publication:

1.      Authors used the term MoS2-M for functionalized MoS2 which is confusing. It is better to use the term F-MoS2.

2.      In the abstract, discuss the quantitative analysis of results.

3.      Please explain the preparation of MoS2 and functionalized MoS2 in detail in the section Materials and Methods.

4.      In Section 2.1 “Functionalized MoS2 (MoS2-M) has been synthesized 67 using this powder (see below).” But below is nothing. Look at this matter.

5.      In section 2.4, “Au electrodes (70 nm thick) were plated on the film by thermal 121 vacuum evaporation (see Fig. 2, left), thus formed two kinds of photodetectors (PDs) 122 based on pure MoS2 and doped-MoS2.” I did not find Fig. 2 left.

6.      Authors should write in the caption that this is the SEM image if functionalized MoS2.

7.      To compare the morphology it is better to provide the SEM images of pure MoS2 and functionalized MoS2 together.

8.      Please add these articles in the introduction part to make it broader.

A.     DOI: 10.1021/acsaelm.3c00851

B.     doi: 10.3390/ma16062247

C.      DOI: 10.1021/acsaelm.2c00567

D.     doi: https://doi.org/10.1002/lpor.202300398

9.      Please confirm the thicknesses of MoS2 and functionalized MoS2 thin films which are used for Raman and photodetector devices by using AFM.

10.   What is the reason for the red-shift in Raman's study?

11.   What is the active area of the devices for photodetectors?

12.   What is the power intensity of light used for photodetection?

13. English language must be improved significantly.

Comments on the Quality of English Language

English language must be improved significantly.

Author Response

Authors are very thankful to reviewers for very valuable comments which helped us to improve the manuscript. All the changes are noticed in red font.

  1. Authors used the term MoS2-M for functionalized MoS2 which is confusing. It is better to use the term F-MoS2.

Answer: Thanks for the suggestive advice. There are no common standards to use such a term. In our study M means molecule, and not chemical element. But F is fluorine atom. So, F-MoS2 is more confusing. That is why we leave our term.

  1. In the abstract, discuss the quantitative analysis of results.

Answer: Thanks for the suggestive advice. Quantitative information has been added to the Abstract.

  1. Please explain the preparation of MoS2 and functionalized MoS2 in detail in the section Materials and Methods.

Answer: Thanks for the suggestive advice. Pure MoS2 nanosheets undergo only ultrasound treatment in solution. We added some additional information and synthesis details of used derivatives to this section.

  1. In Section 2.1 “Functionalized MoS2 (MoS2-M) has been synthesized 67 using this powder (see below).” But below is nothing. Look at this matter.

Answer: Thanks for the suggestive advice. “See below” has been removed.

  1. In section 2.4, “Au electrodes (70 nm thick) were plated on the film by thermal 121 vacuum evaporation (see Fig. 2, left), thus formed two kinds of photodetectors (PDs) 122 based on pure MoS2 and doped-MoS2.” I did not find Fig. 2 left.

Answer: Thanks for the suggestive remark. “see Fig. 2, left” has been removed.

  1. Authors should write in the caption that this is the SEM image if functionalized MoS2.

Answer: Thanks for the suggestive advice. It was added.

  1. To compare the morphology it is better to provide the SEM images of pure MoS2 and functionalized MoS2 together.

Answer: Thanks for the suggestive advice. The morphology of both samples is fully identical. The used molecules are too little, and their amount is not high to change the morphology. It is better to present one bigger picture. Also, these pictures were obtained with different devices with different quality, and we decided to take better picture.

  1. Please add these articles in the introduction part to make it broader.
  2. DOI: 10.1021/acsaelm.3c00851
  3. doi: 10.3390/ma16062247
  4. DOI: 10.1021/acsaelm.2c00567
  5. doi: https://doi.org/10.1002/lpor.202300398

Answer: Thanks for the suggestive advice. Unfortunately, though these papers are interesting, but all of them does not corresponds to our topic.

  1. Please confirm the thicknesses of MoS2 and functionalized MoS2 thin films which are used for Raman and photodetector devices by using AFM.

Answer: Thanks for this helpful advice. The thickness of the films is confirmed by a step profiler (KLA-Tencor), this device already has high accuracy for the thickness test. The thickness of the films used for Raman spectra measurements and microscopy are not determined because they are do not depends on it.

  1. What is the reason for the red-shift in Raman's study?

Answer: Thanks for this careful comment. The presence of red shift of Raman bands means the presence of chemical bonds between MoS2 and molecule, as it is written in the text. Little red shift points out at the fact that only part of the nanosheet surface has been covered with molecules.

  1. What is the active area of the devices for photodetectors?

Answer: The active area of the devices for photodetectors is the region between two Au electrodes separated by 100 μm, i.e., the rectangle area.

  1. What is the power intensity of light used for photodetection?

Answer: Thanks for this careful comment. The incident laser power is 228 μW cm-2, we have enriched this point in the revised version.

  1. English language must be improved significantly.

Answer: Thanks for this careful comment. All the text has been carefully checked by American and English scientists with native English as well as Grammarly program. As you can see the text has undergone numerous changes.

Reviewer 2 Report

Comments and Suggestions for Authors

Dear Editor,

I sincerely thank the editor for the invitation.

This manuscript describes the study of the photodetector based on thick composite films made of multi-layered MoS2 nanosheets chemically bonded with linear molecules of aromatic thiols. The study revealed that the introduction of a paraquaterphenyl thiol derivative effectively boosts both the speed of photoresponse and the photostability of a photodetector based on MoS2 nanosheets. I would recommend this paper for publication only after some minor revisions.

1.     The device structure of the photodetector is depicted in Figure 5 (a-b). It is recommended that the authors include the thickness of the SiO2 layer on the substrate surface, since a too-thin oxide layer may result in a disturbance from the substrate (silicon) signal.

2.     Figure 5(c) shows the I-V curves of the photodetector, and it should be noted that the dark current at negative bias is higher than the photocurrent at different wavelengths of laser irradiation, which indicates that the photodetectors do not have any photo-gain at negative bias, while positive bias does not have this problem. If the absolute logarithmic coordinates are replaced with linear coordinates, the I-V curve will not be straight. So, there is a problem with the ohmic contact between the electrodes and the material. I recommend the authors repair the electrodes before making measurements.

3.     The authors do not explain the increase in device response speed in enough detail and suggest introducing band alignments to explain this phenomenon.

Kind regards,

Comments on the Quality of English Language

The language of this manuscript is generally acceptable for publication in academic journals.

Author Response

Authors are very thankful to reviewers for very valuable comments which helped to improve the manuscript. All the changes are noticed in red font.

  1. The device structure of the photodetector is depicted in Figure 5 (a-b). It is recommended that the authors include the thickness of the SiO2 layer on the substrate surface, since a too-thin oxide layer may result in a disturbance from the substrate (silicon) signal.

Answer: Thanks for this helpful comment. The thickness of SiO2 is 300 nm, which is enough to eliminate the disturbance of bottom Si. We have pointed out the thickness of SiO2 layer in revised file to address this issue.

  1. Figure 5(c) shows the I-V curves of the photodetector, and it should be noted that the dark current at negative bias is higher than the photocurrent at different wavelengths of laser irradiation, which indicates that the photodetectors do not have any photo-gain at negative bias, while positive bias does not have this problem. If the absolute logarithmic coordinates are replaced with linear coordinates, the I-V curve will not be straight. So, there is a problem with the ohmic contact between the electrodes and the material. I recommend the authors repair the electrodes before making measurements.

Answer: Thanks for this careful review. The reason for this phenomenon can be the formation of Schottky contacts between metal and semiconductor. In the case of Schottky contact, the band of semiconductor at the interface is curved, thus forming a Schottky barrier, showing a nonlinear I-V behavior under positive/negative bias. So, it’s rational that the photocurrent is higher than dark current at positive bias and lower at negative bias because of the existence of built-in electric field. We will take it into account in our future experiments.

  1. The authors do not explain the increase in device response speed in enough detail and suggest introducing band alignments to explain this phenomenon.

Answer: Thanks for this helpful comment. The observed response increase is explained by better connection between nanosheets owing to functionalization with proposed molecule. These are our first results, and further studies are needed.

Reviewer 3 Report

Comments and Suggestions for Authors

In this manuscript, Kukhta. et al. solution processed functionalized MoS2 thick composite layer is employed for photodetection application. The manuscript does not provide any significant results, so that the manuscript can be considered for publication in MDPI Photonics. Major manuscript revision and improvement shall be carried out before consideration. My detailed comments are as follows:

Abstract is lacking quantitative results; it appears that it’s a general introduction.

Authors should emphasize the advantages of chemical methods, while there are standard CVD and even large area CVD growths report are in literature.

Furthermore, even the photodetector response is not comparable to the mechanically exfoliated MoS2.

The functionalization is just enhancing the current by less than 1 order of magnitude, (fig. 5).

What is the type of polarity of the obtained and functionalized MoS2?

Authors should clarify whether the article is mainly focused on synthesis approach of MoS2 and functionalized MoS2 or photodetector application.

More figures of merits, such as responsivity, detectivity, sensitivity, etc. should be calculated and inserted with comparison to the existing literature.

What were the experimental conditions during photodetection measurements, such as Vgs, chamber pressure?

Role of surface adsorbates over the channel should be discussed ?

What was the incident laser power over the fabricated device?

Exponential rise/decay current fit should be shown.

Reviewer suggest that previous works and reviews on other MoS2 and functionalized MoS2 shall be added in the Introduction section, with experimental data mentioning which could provide a more comprehensive view in the field, such as: Phys. Status Solidi A 2300107; Materials Today Nano 24, 100382; Nanomaterials 2023, 13(9), 1491; Journal of Physics and Chemistry of Solids 179, 111406; Adv. Sci. 2023, 10, 2207743.

Conclusions should be supported with the experimental results?

Author Response

Authors are very thankful to reviewers for very valuable comments which helped us to improve the manuscript. All the changes are noticed in red font.

Abstract is lacking quantitative results; it appears that it’s a general introduction.

Anwser: Thanks for this helpful comment. Quantitative results have been added.

Authors should emphasize the advantages of chemical methods, while there are standard CVD and even large area CVD growths report are in literature.

Anwser: Thanks for this helpful comment. Chemical methods are much simpler, and there is no other possibility to make functionalization with large organic molecules except chemical methods.

Furthermore, even the photodetector response is not comparable to the mechanically exfoliated MoS2. The functionalization is just enhancing the current by less than 1 order of magnitude, (fig. 5).

Answer: Thanks for this constructive comment. There is no denying that the device performance is poor in this paper, photodetectors based on mechanically exfoliated MoS2 has much better performance, which has been reported so far. The innovation points of this paper are the relative performance improvements after functional organic molecule doping, which hasn’t been reported so far, thus having certain significance for theoretical research and practical applications. And we are convinced that further development of this approach can help to reach an excellent performance.

What is the type of polarity of the obtained and functionalized MoS2?

Answer: Thanks for this helpful comment. Pristine molybdenum disulfide (MoS2) monolayer demonstrates predominant and persistent n-type semiconducting polarity due to the natural sulfur vacancy. Functionalization with p-type organic molecule can change MoS2 polarity depending on the degree of functionalization and molecule properties.

Authors should clarify whether the article is mainly focused on synthesis approach of MoS2 and functionalized MoS2 or photodetector application.

Anwser: Thanks for this helpful comment. The new material proposed for successful formation of solution processing thin films and PD ability is considered. These are our first results, and further studies are needed.

More figures of merits, such as responsivity, detectivity, sensitivity, etc. should be calculated and inserted with comparison to the existing literature.

Answer: Thanks for this constructive suggestion. The calculated responsivity, detectivity and sensitivity of our device with proposed approach aren’t comparable to previous reports, thus it’s unnecessary to list these parameters and compare them with other existing literatures. What we want to show is that the photoresponse and photo-stability do get improved after functional organic molecule doping.

What were the experimental conditions during photodetection measurements, such as Vgs, chamber pressure?

Answer: Thanks for this careful comment. The test conditions of photodetection measurements are ambient environment, i.e., room temperature and atmospheric pressure. And the photodetectors we constructed is a terminal device, which has no Vgs.

Role of surface adsorbates over the channel should be discussed ?

Answer: Thanks for this careful comment. Because all the measurements were made at ambient conditions, atmospheric substances such as oxygen and water vapors can be adsorbed by active material and can influence the results. However, this study represents first steps in this direction and the influence of adsorbates is not considered.

What was the incident laser power over the fabricated device?

Answer:  Thanks for this careful comment. The incident laser power is 228 μW cm-2, we have enriched this point in the revised version.

Exponential rise/decay current fit should be shown.

Answer: Thanks for this comment. We have shown the semilogarithmic I-t curves of devices in Figure 6, in which the rise/decay current can be seen. In fact, there are two or more decay times, fast and slow. We consider here only fast component. It is linear, i.e. exponential.

Reviewer suggest that previous works and reviews on other MoS2 and functionalized MoS2 shall be added in the Introduction section, with experimental data mentioning which could provide a more comprehensive view in the field, such as: Phys. Status Solidi A 2300107; Materials Today Nano 24, 100382; Nanomaterials 2023, 13(9), 1491; Journal of Physics and Chemistry of Solids 179, 111406; Adv. Sci. 2023, 10, 2207743.

Answer: Thanks for this comment. We added these papers to the reference list, except the forth which does not correspond to the considered topic.

Conclusions should be supported with the experimental results?

Answer: Thanks for this correct review. Of course, the conclusions should be supported with the experimental results. We have comprehensively analyzed the obtained data and drawn rational conclusions based on experimental results. All the results are in good agreement with each other.

Round 2

Reviewer 1 Report

Comments and Suggestions for Authors

The authors did not respond to the comments properly. This manuscript lacks a few major things that hinder its acceptance as:

1. The SEM is not provided for pure MoS2.

2. Thickness can play an important role in Raman's study so authors failed to provide it.

3. The dimensions (Width and thickness) of the device are also not clear. 

Author Response

Authors are very thankful for this careful review.

Does the introduction provide sufficient background and include all relevant references?

Response: We checked again the background and all the available literature carefully.

Are all the cited references relevant to the research?

Response: We checked all the available literature carefully.

Is the research design appropriate?

Response: We added a picture of pure MoS2 sample, an additional information to Raman’s study, and the dimensions of the active area.

Are the methods adequately described?

Response: We added the second microscope used in the measurements.

Are the results clearly presented?

Response: We added a picture of pure MoS2 sample.

Are the conclusions supported by the results?

Response: We added a picture of pure MoS2 sample and an additional information to Raman’s study, to confirm the conclusions.

  1. The SEM is not provided for pure MoS2.

Response: Thank you for pointing this out. The SEM picture for pure MoS2 was added (see page 5). It was a good idea, though these pictures were obtained with different devices and with different degree of sonication. They confirm the description and conclusions.

  1. Thickness can play an important role in Raman's study so authors failed to provide it.

Response: Thank you for pointing this out. We agree that the thickness can play an important role in Raman’s study. There are a number of studies of single layer and several layers independent nanosheets. It was found that the position and ratio of the main Raman bands depend on the number of layers up to several layers. But, there is a very scarce information about Raman spectra in the composites where nanosheets with different dimensions and thickness are distributed chaotically. We found the only paper with similar studies: Domashevskaya et al, Semiconductors, v. 53, Issue 7, 2019 (in Russian). According to this paper, the E12g band is structure thickness sensitive. In our case the thickness can be 1.3 μm, because all the films are prepared with the same tool and the same amount and concentration of the dispersion. Such thickness can be considered as a bulk material, and our measurements of position and bands ratio corresponds to the bulk data. Moreover, for both samples we made measurements at five points of the samples. They include different nanosheet dimensions, distribution, and morphology. These measurements showed that the band intensities were slightly different, but position, ratio, and bandwidth were the same at the every point. Our data confirm the functionalization of MoS2 nanosheets. Though, this question requires independent studies. This information has been added to the text (see page 6). The functionalization is confirmed also by optical, luminescence, and microscopy data.

  1. The dimensions (Width and thickness) of the device are also not clear. 

Response: Thank you for pointing this out. According to the electron microscopy data, the electrode width is 500 μm, the gap is 100 μm, film thickness is 1.3 μm, SiO2 film thickness is 300 nm. The width value has been added to the text (see page 7).

Reviewer 3 Report

Comments and Suggestions for Authors

Authors have addressed all the comments and the revised manuscript can be accepted for publication in the present form.

Author Response

Authors are very thankful to reviewrs for this very careful review. It helped to make our manuscript better.
